# Ambient air pollution exposure and effects on neutralizing antibody titers following SARS-CoV-2 vaccination in adults

Daniel P. Croft[1,2]*, Carl J. Johnston[3], Angela R. Branche[4], David Q. Rich[1,2,5], Philip K. Hopke[2,5,6], Kelly Thevenet-Morrison[5], Sally W. Thurston[2,7], Todd A. Jusko[2,3,5], Md Rayhanul Islam[5], Catherine Bunce[4], Michael C. Keefer[4], Edward E. Walsh[4], Ann R. Falsey[4]

1 Department of Medicine, Division of Pulmonary and Critical Care Medicine, University of Rochester, Rochester, New York, United States of America, 2 Department of Environmental Medicine, University of Rochester Medical Center, Rochester, New York, United States of America, 3 Department of Pediatrics, University of Rochester. Rochester, New York, United States of America, 4 Division of Infectious Disease, Department of Medicine, University of Rochester, Rochester, New York, United States of America, 5 Department of Public Health Sciences, University of Rochester, Rochester, New York, United States of America, 6 Institute for a Sustainable Environment, Clarkson University, Potsdam, New York, United States of America, 7 Department of Biostatistics and Computational Biology, University of Rochester, Rochester, New York, United States of America

* daniel_croft@urmc.rochester.edu

## Abstract

We examined the association between air pollution and neutralizing antibody responses to COVID-19 vaccination in participants enrolled in a phase 3 clinical trial. Seventy-four adults were vaccinated with two doses of the AstraZeneca ChAdOx1 vectored vaccine (AZD1222) ($5 \times 10^{10}$ viral particles) at baseline and day 29, between Aug 28, 2020, to Jan 15, 2021, in Monroe County, NY. SARS-CoV-2 pseudovirus neutralizing $ID_{50}$ titers (NAb) and total spike protein IgG were assessed at baseline and 15, 29, 43, 57 and 90 days after vaccination. In this pilot study, each participant's dates of neutralization titers were matched to Monroe County ambient concentrations of fine particles ($PM_{2.5}$; ≤ 2.5 μm), black carbon (BC; marker of traffic), among other particulate and gaseous pollutants. Using linear mixed models, we estimated the association between each interquartile range (IQR) difference in air pollutant concentrations in the 14 days prior to blood collection and antibody responses at each post vaccination timepoint. Though not statistically significant, we observed a 23% reduction in NAb titer (95% CI: -67%, 79%) measured on day 43 (i.e., 14 days after second vaccination) associated with each 0.32 μg/m³ increase in BC concentrations in the prior 14 days. We also observed a 42% increase in spike protein IgG (95% CI: -16%, 141%) measured on day 15 (i.e., 14 days after primary vaccination) associated with each 0.26 μg/m³ increase in BC concentrations in the 14 days prior. A similar pattern for total spike protein IgG was observed at day 29 (42%; 95% CI: -22%, 157%) and 90 (43%; 95% CI: -11%, 127%). Future research will need to explore the possible

**Data availability statement:** Data underlying the findings described in this manuscript may be obtained in accordance with AstraZeneca's data sharing policy described at https://astrazenecagrouptrials.pharmacm.com/ST/Submission/Disclosure.? Data for studies directly listed on Vivli (which includes the AZ COVID trial used in this study) can be requested through Vivli at www.vivli.org. AstraZeneca Vivli member page is also available outlining further details: https://vivli.org/ourmember/astrazeneca/.

**Funding:** This work was supported by a NIEHS Research Career Development Award [K23 ES032459 to DPC] and NIH P30 ES001247 [to the Environmental Health Science Center of which DPC, DQR, PKH and TJJ are members]. The entire project was also supported by the National Institute of Allergy and Infectious Diseases (NIAID) VTEU grant [5UM1AI148450-02 to ARF and ARB]. The funders had no role in study design, data collection and analysis, decision to publish, or preparation of the manuscript.

**Competing interests:** Dr. Angela Branche reports financial support including research funding from Moderna, Pfizer, Cyanvac and Vaccine.com. Dr. Branche has served on an advisory board for Novavax and speakers bureau for Moderna and GSK. Dr. Ann R. Falsey has received consulting fees and research grants from Merck Sharpe and Dohme, Pfizer, Astra Zeneca, Novavax, Janssen, CyanVac, Vax Co, Moderna, Sanofi Pasteur, GSK and BioFire Diagnostics. Dr. Edward E. Walsh reports grants from Merck and Pfizer Inc, served as a consultant to Moderna, Merck, GlaxoSmithKline, and Janssen. If there are other authors, they declare that they have no known competing financial interests or personal relationships that could have appeared to influence the work reported in this paper.

association between air pollution exposure and antibody response to SARS-CoV-2 vaccination given the potential for compromised vaccine efficacy.

## Author summary

The majority of the world breathes outdoor $PM_{2.5}$ concentrations above the WHO standard. COVID-19 vaccines protect individuals by causing an immune response leading to increased Neutralizing antibodies (NAb) to the spike protein, a selected antigen from the SARS-CoV-2 virus. Past studies have observed changes in the immune response to vaccines related to environmental exposures. In our pilot study we matched the 14 days of pollution prior to each blood draw in a group of 74 patients who received a specific COVID-19 vaccine (AZD1222). In part due to our small sample size, our results did not reach statistical significance and had high variability. Though imprecise, we observed that exposure to $PM_{2.5}$ and other pollutants generally showed a pattern of decreased NAb titers in the time period after the second AZD1222 vaccine. The fact that Rochester, NY is a low pollution area makes our study policy relevant as our $PM_{2.5}$ concentrations are below the National Ambient Air Quality Standards. While our study will need to be replicated in a larger multicenter study, if air pollution reduces immunity from AZD1222, vaccine dosing or frequency may need adjustment to maximize efficacy according to local pollution exposure.

## Introduction

In December 2019, a novel β-coronavirus (SARS-CoV-2) was identified in China as a cause of severe pneumonia (COVID-19) with a high degree of human-to-human transmission [1,2]. Beginning in the winter of 2020, COVID-19 vaccines were recommended as a means of primary prevention [3]. However, optimal responses to vaccinations are dependent on intrinsic host factors, medication use and environmental factors (including toxicant exposures) [4].

Environmental exposures such as heavy metal pollution and persistent organic pollutants are known to attenuate immune responses to multiple vaccinations[5–7]. Other environmental exposures, such as air pollution, can also impair the human immune system [8,9], with recent epidemiologic studies documenting an increased risk of respiratory viral infection in adults with higher exposures to $PM_{2.5}$ (particulate matter ≤2.5μm in diameter) [10–12]. Multiple mechanisms are suspected to contribute to this increased risk of infection including innate immune dysfunction, impaired gene expression and impaired antiviral immune cell signaling [13]. However, the association between air pollution and SARS-CoV-2 vaccine response is relatively understudied. A 2022 study of 207 adults in China documented a 3.4 AU/ml (31%) lower neutralizing antibody titers (NAb) to an inactivated SARS-CoV-2 vaccine (Vero cell, CoronaVac,

SINOVAC, China) associated with a modelled increase in the mean composite air pollution concentration over the past year [14]. Another study in Spain reported a 10% decrease in antibody response to spike protein associated with interquartile range (IQR) increases in $PM_{2.5}$, $NO_2$ and Black carbon [15]. Both studies (China and Spain) were conducted in higher pollution settings (24hr average $PM_{2.5}$ of 30 μg/m³ and 16 μg/m³ respectively) than is common in the U.S. and also used long term pollution data (1 year or more prior to vaccination or infection respectively). The question remains as to whether this effect on vaccine response is associated with acute increases in air pollution in a low air pollution area (annual average [$PM_{2.5}$] of ≤ 12μm/m³); a concentration commonly found in the United States and other parts of the world. With 99% of the world population breathing unhealthy air [16] and 64% of the world receiving at least one COVID-19 vaccine [17], a better understanding of the association between air pollution and vaccine responses is urgently needed.

The present pilot study, in Western New York leveraged an existing trial of Astra Zeneca (AZ) Phase 3 trial of the ChAdOx1 vectored COVID vaccine (AZD1222) [18]. This study included several outcome measures including pseudovirus NAb or total spike protein IgG antibody responses at multiple days after primary and secondary vaccination. We estimated the association between interquartile range increases in ambient air pollution in the previous 14 days and serial measures of antibody response longitudinally for 90 days after primary or secondary vaccination, hypothesizing that exposure to increased PM concentrations would be associated with an attenuated antibody response after vaccination.

## Methods

### Ethics statement

The clinical trial from which this post-hoc analysis enrolled participants was approved by the Advarra central institutional review board (IRB). The University of Rochester Research Subjects Review Board (local IRB), approved this study protocol (#7384) for a post-hoc analysis of the clinical trial. Formal consent was obtained for all participants except for participants who were lost to follow up from the clinical trial upon which this post-hoc analysis was based (waiver of consent granted).

### Study population

The University of Rochester was a member of the BARDA-funded US COVID-19 Prevention Network (COVPN), which participated in the Astra Zeneca (AZ) Phase 3 trial of the ChAdOx1 vectored COVID vaccine (AZD1222) [18]. From August 28, 2020, to January 15, 2021, 97 adults (≥18 years of age) were enrolled in the phase 3 clinical trial of the AZ COVID19 vaccine in Rochester, NY (NCT04516746). As part of the trial each participant was vaccinated with AZD1222 (5 x 10¹⁰ viral particles) at days 1 and 29 of the vaccine trial (the actual day 1 differed between participants based on date of enrollment into trial). Ninety-seven participants who were included in the immunogenicity subset of the Astra Zeneca vaccine trial had serial measurements of pseudovirus NAb (Inhibitory dilution [50%]; $ID_{50}$) and a spike protein IgG at baseline (Day 1) and days 15, 29, 43, 57 and 90 after vaccination [Day 90 only for spike protein IgG]) (Fig 1). As this present study focused on the effect of air pollution on immune response to vaccination, only participants who received the AZD1222 vaccine were included (not individuals in the placebo arm). Also, due to the restriction of the sub-study population to participants living within 15 miles of the central air pollution monitor, and a small number of missing pollution values, only 74 participants were included in the analysis.

### Data collection

To ascertain additional information for the present study of air pollution and vaccination response, participants were asked additional questions during the consent process that included questions about smoking behaviors and environmental tobacco smoke, types of cooking stoves used in the home, and the number of hours outside the home each day. We also used data collected during the previous trial including demographic information (participant race/ethnicity, age, and sex) and health information including pertinent comorbidities (e.g., BMI [calculated using clinic measure height and weight], diabetes, asthma, COPD and immunosuppression) and medication use (e.g., inhaled or oral steroid use, statin use).

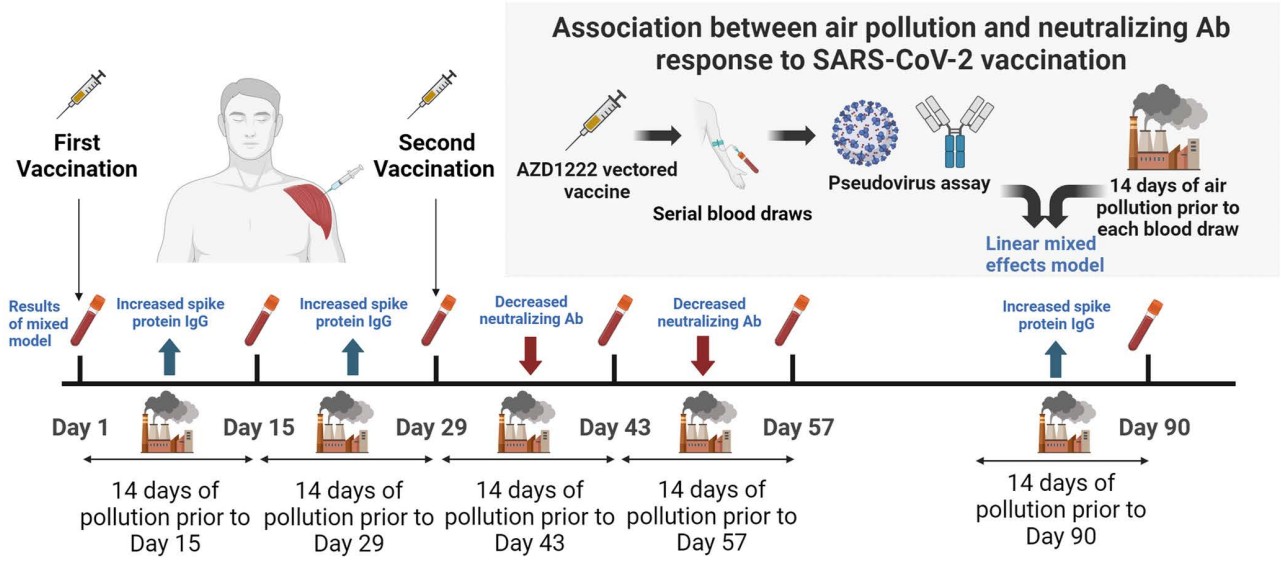

**Fig 1. Design and results of the pilot secondary analysis of the AZD1222 clinical trial.**

## Air pollution data collection

Rochester, NY is a relatively low pollution area (e.g., average $PM_{2.5}$ of 5–8 µg/m³) [19]. Daily measured concentrations of $PM_{2.5}$, black carbon (BC; marker of traffic pollution), Delta-C (DC; a marker of wood smoke), ultrafine particles (UFP), accumulation mode particles (AMP), Nitrogen dioxide ($NO_2$), Sulfur dioxide ($SO_2$), Carbon monoxide (CO), and ozone ($O_3$) were collected in the 14 days prior to a blood draw timepoint in the vaccine trial in Monroe County, NY.

Pollution measurements of the particle number counts (PNC) in the size range of 10–500 nm were made continuously and sequentially at a New York State Department of Environmental Conservation (NYS DEC) site in Rochester, NY [20]. Briefly, measurements were made at the NYS DEC primary site on the eastside of Rochester, NY at the intersection of two major interstate highways (I-490 and I-590) and a local NY route 96. Hourly $PM_{2.5}$ mass and gaseous pollutant concentrations, ambient temperature and relative humidity, wind speed and direction, were also measured at the same intersection of highways. Size distribution measurements were made using a scanning mobility particle sizer (SMPS, TSI Inc.) system. The size range bounds were 10.4 nm (lower) and 0.542 µm (upper). $PM_{2.5}$ was measured with a TEOM (model 1400ab, Thermo Fisher Scientific Inc., USA), and Black Carbon was measured with a 2-wavelength aethalometer (Magee Scientific, Inc., USA). The pollutant gases measured with standard FEM gas monitors (Thermo Fisher Scientific, Inc., USA).

## Serum spike protein IgG

Serum IgG responses to the SARS-CoV-2 spike protein were measured using a validated quantitative multiplexed electrochemiluminescent assay (PPD Vaccines, Richmond, VA). Antibody concentrations were determined using an indirect binding assay on the Meso Scale Discovery platform. A reference standard was established by combining pre-screened Covid-19–positive human serum samples. Antibody concentrations were determined by interpolating relative light units to a standard curve generated from the serially diluted reference standard and assigned a concentration in arbitrary units (AU)/mL.

## Serum neutralizing antibody

NAb were measured using a validated lentivirus-based SARS-CoV-2 pseudovirus assay (Monogram Biosciences, South San Francisco, CA). Serial dilutions of serum were preincubated with pseudovirions containing luciferase and a Wuhan-Hu-1

spike protein. Antibody titers were reported as the reciprocal of the serum dilution conferring 50% inhibition (ID50) of pseudovirus infection. A control containing a non-specific pseudovirus was utilized to determine activity specific to SARS-CoV-2. The titer was then transformed to the log 2 or log 4 depending upon the dilutional factor. For example, if a titer started at 1:50 and the 3rd 2-fold dilution from there (1:400) neutralizes 50% of the pseudovirus but the 1:800 does not, the titer is calculated at log2 of 1:400 which is 8.64. These unitless titer values served as the titer values in the present study.

## Statistical analysis

To examine the association between the NAb response to **primary vaccination** on Day 1 and short-term increases in ambient air pollutant concentration, linear mixed effects models were used. All NAb titers were transformed by log base 2 to meet normality assumptions. The serum titers of NAb from days 15 and 29 were regressed on the mean $PM_{2.5}$ concentration in the previous 14 days, in a linear mixed effects model that included a random subject effect, and adjusted for participant sex, an indicator variable for day (15 or 29), and an interaction term between day and $PM_{2.5}$.

The association between the NAb response to the **second vaccination** on day 29 and $PM_{2.5}$ in the previous 14 days was similarly examined. The serum NAb titers from days 43 and 57 were regressed on the mean $PM_{2.5}$ concentration in the previous 14 days in a model with a random subject effect, adjusting for sex, an indicator variable for day (43 or 57), an interaction term between day and $PM_{2.5}$, and the actual day of the blood draw minus the expected date of blood draw (e.g., if a blood draw was at day 46 and expected at day 43, the value would be 3). Both sets of models were rerun with spike protein IgG titer as the outcome in the same manner, where the second model included outcomes at days 43, 57, and 90, and the indicator variables for day (43, 57, or 90) and interaction between day and $PM_{2.5}$ were expanded accordingly. These models for spike protein IgG were adjusted for age (continuous variable) instead of sex. While all models included the day and interaction of day and pollutant, only sex and age were included as covariates in the models for NAb and spike protein IgG (respectively) as these variables were significant predictors of the outcome. BMI, smoking, gas stove use, comorbidities (including COPD, asthma, diabetes), humidity and temperature were not included as significant predictors of either outcome. Separate models with other air pollutants (BC, AMP, DC, $SO_2$, $NO_2$, $O_3$) were fit in a similar manner. From each model, the percent difference in NAb or spike protein IgG associated with each interquartile range (IQR) increase in pollutant concentration, and its 95% confidence interval were estimated. In this case, IQR was chosen as the unit of modelled increase for the regression as it ensures a proportional increase in each pollutant to allow for comparison between pollutants. (S1 Text). Of note, the linear regression method proc glmselect with stepwise selection was used for covariate selection, keeping only those covariates that were significant predictors of the outcome ($p < 0.05$). The variables that we determined to be significantly associated with the outcomes were those covariates with the highest frequency of significant associations in these 90 models of multiple time points and multiple pollutants. Sex was the variable that was observed most frequently for models with NAb, and age was observed most frequently for models with spike protein IgG.

Lastly, to examine if the antibody responses were the same for those with and without indoor $NO_2$ exposure, the median NAb and spike protein IgG in participants who cook with gas stoves compared with those with electric stoves were calculated. All analyses were conducted using SAS 9.4 (Cary, NC). While statistical significance was defined as a p value <0.05, we focused on the direction and magnitude of the effect estimates in this study to determine whether a consistent effect was observed between immune response outcomes (NAb and spike protein IgG) and air pollution. While the p values for effect estimates are not presented, the presence of non-statistically significant findings can be inferred by the confidence intervals crossing zero.

## Results

### Demographic and clinical characteristics of study cohort

The majority of participants reported White race (82%) and male sex (69%) (Table 1). The average age was 60 years old with a standard deviation of 17 years. The most frequently reported medical comorbidities included asthma (18%)

**Table 1. Demographic characteristics of study population from Monroe County, NY in 2020-2021.**

| | N=74 |
|---|---|
| **Female Sex, No. (%)** | 23 (31.1) |
| **Mean Age, year (SD)** | 59.5 (17.1) |
| **Mean BMI, kg/m² (SD)** | 29.7 (6.6) |
| **Mean Time (hrs.) spent outside of home each day (SD)** | 3.6 (3.4) |
| **Race/Ethnicity, No. (%)** | |
| White | 61 (82.4) |
| Black/African American | 7 (9.5) |
| Asian | 1 (1.4) |
| Multiple | 5 (6.8) |
| Hispanic or Latino | 9 (12.2) |
| **Baseline Medical and Exposure History, No. (%)** | |
| Immunosuppressed | 2 (2.7) |
| Diabetes mellitus | 8 (10.8) |
| Congestive heart failure | 3 (4.1) |
| Chronic obstructive pulmonary disease (COPD) | 1 (1.4) |
| Asthma | 13 (17.6) |
| Chronic renal failure | 1 (1.4) |
| Active Smoker | 3 (4.3) |
| History of Tobacco Smoking | 6 (8.1) |
| Use of Oral steroid | 2 (2.7) |
| Use of Inhaled steroid | 8 (10.8) |
| Use of Statin medication | 21 (28.4) |
| Gas stove use in dwelling | 37 (53.6) |
| Use of Home Oxygen | 0 (0) |

and diabetes (11%), followed by congestive heart failure (4%) and immunosuppression (3%). Only 1% or fewer reported COPD or chronic renal failure. Three percent of participants reported current oral steroid use and 28% were on statins. Eight percent of participants actively smoked and 54% used gas stoves for cooking in their home.

### Distributions of air pollution and antibody measurements

The concentrations of air pollutants measured for the six different 14-day times periods was stable with no consistent patterns of increase or decrease when comparing time points (Table 2). As the study went from August to January, there was only one seasonal transition during the period (Fall to Winter). Similar to other cities in the Northeastern United Status, wintertime in Rochester, NY is generally a time of high $NO_2$ concentrations and low ozone concentrations[21].

### Air pollutants and neutralizing antibody

There was no clear trend in the percent changes in NAb at days 15 or 29 associated with IQR increases in pollutants in the 2 weeks prior to NAb measurement (Table 4A; S1A Fig). Although not statistically significant, we observed reductions in NAb associated with most pollutants at days 43 and 57 (Table 4A; S1B Fig). Decreases of 13% (95% CI: -52, 60) and 14% (95% CI: -60, 85) in NAb titers were associated with an IQR increase in $PM_{2.5}$ on days 43 (2.3 μg/m³) and 57 (2.1 μg/m³), respectively (Table 4A). A similar magnitude of reduction in NAb was observed for 0.3 μg/m³ increases in black carbon in the 14 days prior to Day 43 (-23% [95% CI: -67, 79]) and Day 57 timepoints (-14% [95% CI: -60, 83]). Though there

Table 2. Air pollution concentrations in the 14 days prior to each blood draw timepoint.

| | Pollutant | Day 15 Median (IQR) | Min, Max | Day 29 Median (IQR) | Min, Max | Day 43 Median (IQR) | Min, Max | Day 57 Median (IQR) | Min, Max | Day 90 Median (IQR) | Min, Max |
|---|---|---|---|---|---|---|---|---|---|---|---|
| Spike protein IgG (N=74) | $PM_{2.5}$ | 5.8 (1.9) | 2.9, 11.5 | 6.2 (2.0) | 2.5, 12.8 | 6.0 (2.1) | 2.5, 12.8 | 5.3 (2.9) | 2.9, 8.1 | 6.8 (2.7) | 4.3, 11.1 |
| | BC | 0.3 (0.3) | 0.1, 0.8 | 0.4 (0.3) | 0.1, 1.0 | 0.4 (0.3) | 0.1, 1.0 | 0.4 (0.3) | 0.2, 0.5 | 0.4 (0.3) | 0.2, 0.6 |
| | DC | 0.2 (0.1) | 0.0, 0.4 | 0.2 (0.1) | 0.0, 0.5 | 0.2 (0.1) | 0.0, 0.5 | 0.1 (0.2) | 0.0, 0.3 | 0.2 (0.1) | 0.1, 0.3 |
| NAb (N=74) | UFP | 4,360.0 (3,312.7) | 1,691.1, 7,559.4 | 4,092.0 (2,983.6) | 1,560.4, 8,747.4 | 4,360.0 (3,520.2) | 1,432.0, 9,837.0 | 4,655.2 (2,951.5) | 1,560.4, 9,431.7 | 4950.3 (6244.1) | 2088.0, 8747.4 |
| | AMP | 548.8 (341.0) | 147.2, 1,318.9 | 419.2 (375.0) | 151.1, 1,464.8 | 540.7 (445.3) | 133.0, 1,464.8 | 437.2 (386.2) | 148.8, 1,258.9 | 419.2 (333.5) | 208.6, 1326.2 |
| | $NO_2$ | 7.2 (3.1) | 3.1, 13.3 | 7.1 (3.8) | 2.6, 15.6 | 7.0 (3.8) | 2.6, 15.6 | 5.7 (3.4) | 3.1, 9.8 | 7.1 (2.5) | 4.6, 9.5 |
| | $O_3$ | 0.03 (0.01) | 0.01, 0.04 | 0.03 (0.01) | 0.01, 0.04 | 0.03 (0.01) | 0.01, 0.04 | 0.03 (0.01) | 0.01, 0.04 | 0.03 (0.01) | 0.01, 0.04 |
| | CO | 0.2 (0.1) | 0.1, 0.4 | 0.2 (0.1) | 0.1, 0.4 | 0.2 (0.1) | 0.1, 0.4 | 0.2 (0.1) | 0.1, 0.3 | 0.2 (0.1) | 0.2, 0.3 |
| | $SO_2$ | 0.2 (0.4) | 0.0, 0.8 | 0.2 (0.5) | 0.1, 0.9 | 0.2 (0.3) | 0.0, 0.9 | 0.2 (0.1) | 0.0, 0.9 | 0.2 (0.2) | 0.1, 0.6 |

$PM_{2.5}$, BC and DC are measured in $\mu g/m^3$, UFP and AMP in particles/$cm^3$, $NO_2$, $SO_2$ and CO in ppb and $O_3$ in ppm.

Mean Neutralizing Ab titers and spike protein IgG concentrations increased after vaccination with peak responses noted at Day 43, two weeks after the second vaccine (Table 3). The variability of NAb was largest at day 43, while the variability of spike protein IgG was largest at Day 29 and 43.

**Table 3. Distribution for both log2 transformed and actual Neutralizing Ab titers and Spike protein IgG concentrations by lab draw day.**

| | Day 15 | | Day 29 | | Day 43 | | Day 57 | | Day 90 | |
|---|---|---|---|---|---|---|---|---|---|---|
| | Mean (SD) | Min, Max | Mean (SD) | Min, Max | Mean (SD) | Min, Max | Mean (SD) | Min, Max | Mean (SD) | Min, Max |
| **Log-transformed value** | | | | | | | | | | |
| **Spike protein IgG** | 11.2 (3.2) | 4, 14.7 | 12.3 (3.2) | 5, 17.3 | 14.6 (2) | 7.5, 17.7 | 14.1 (1.8) | 8.5, 17.2 | 13.8 (2.3) | 9.1, 19.4 |
| **Neutralizing antibody** | 4.3 (1.7) | 4.3, 9.7 | 4.3 (1.3) | 4.3, 10.5 | 7.7 (2.9) | 4.3, 11.6 | 7.6 (2.1) | 4.3, 11.5 | | |
| **Untransformed value** | | | | | | | | | | |
| **Spike protein IgG (AU/ml)** | 2428 (4902) | 17, 25897 | 5158 (10851) | 33, 157495 | 24651 (29866) | 186, 209204 | 18132 (21620) | 352, 148303 | 14727 (21749) | 548, 672912 |
| **Neutralizing antibody** | 20 (43) | 20, 815 | 20 (28) | 20, 1457 | 203 (318) | 20, 3009 | 194 (325) | 20, 2982 | | |

were different patterns of effect when comparing the period after the first vaccination (Days 15 and 29) and the period after the second vaccination (Days 43 and 57), there were no statistically significant differences between the estimates after the first vaccination and the estimates after the second vaccination for any pollutant.

## Spike protein IgG responses to changes in pollutant IQR

We observed an increase in Spike protein IgG associated with an IQR increase in pollutant concentrations at all timepoints (Table 4B; S2 Fig). Similar to the neutralizing titers, the effects of pollution increases were not statistically significant, but an increase in antibody levels was observed in association with increased concentrations of most of the specific pollutants measured. Specifically, we observed a 25% (95% CI: -4%, 63%) and 33% (95% CI: -7, 89) percent increase in spike protein IgG, associated with IQR increases in $PM_{2.5}$ at days 15 (1.9 µg/m³) and 29 (2.0 µg/m³), respectively (Table 4B). Compared to $PM_{2.5}$, larger percent differences were observed for the association between spike protein IgG at Day 15 and 29 and 0.3 µg/m³ increases for black carbon (42% [95% CI: -16, 141] and 42% [95% CI: -22, 157] respectively). After the second vaccination on day 29, similar changes in spike protein IgG were observed at the day 43, 57 and 90 timepoints associated with IQR increases in pollution. After the second vaccine the largest magnitude of percent increases were observed at day 90 for $PM_{2.5}$ and BC, compared with lower percent changes at day 43, and 57. For example, a 2.4 µg/m³ increase in $PM_{2.5}$ concentrations in the 14 days prior to the day 90 timepoint was associated with a spike protein IgG increase of 35% (95% CI: 2, 78), compared to a -6% (95% CI: -28, 24) and 16% (95% CI: -22, 71) change at day 43 (2.1 µg/m³ increase) and 57 (2.9 µg/m³ increase), respectively. The percent change in spike protein IgG associated with IQR increases in black carbon (day 90: 0.1 µg/m³, day 43: 0.3 µg/m³) was also larger at day 90 (43% [95% CI: -11, 127]), than at Day 43 (21% [95% CI: -28, 103]). Broadly, the other particulates (DC, UFP and AMP) and gasses $NO_2$ and CO followed a similar pattern. In contrast, we observed reductions in spike protein IgG associated with IQR increases in Ozone ($O_3$) and an inconsistent pattern for $SO_2$.

## The relationship of antibody responses to indoor air quality

In a simple descriptive sensitivity analysis, we explored whether median antibody titers/concentrations were different in individuals with or without gas stoves in their homes. For simplicity, we explored this gas stove/median antibody association independent of ambient air pollution. Though the log transformed value for neutralizing Ab titers appeared slightly lower (10%) at the day 57 lab draw for participants living in a home with a gas stove compared to those who did not live in a home with a gas stove, this change was not observed at other time points. There were no consistent changes observed for spike protein IgG concentrations when considering gas stove use. Overall, there did not appear to be a consistent

Table 4. Percent change (and 95% CI) in (A) neutralizing antibody titers and (B) spike protein IgG concentrations at multiple timepoints associated with an IQR increase in multiple pollutants in the 14 days prior to each timepoint.

| Pollutant Days (Blood draw day) | | Day 1–14 (Day 15 draw) | | Day 15–28 (Day 29 draw) | | Day 29–42 (Day 43 draw) | | Day 43–56 (Day 57 draw) | | Day 77–90 (Day 90 draw) | |
|---|---|---|---|---|---|---|---|---|---|---|---|
| | | Percent difference | 95% CI | Percent difference | 95% CI | Percent difference | 95% CI | Percent difference | 95% CI | Percent difference | 95% CI |
| A. Neutralizing Ab (N=74) | $PM_{2.5}$ | 4.4 | (-18.8, 34.2) | -6.9 | (-31.9, 27.3) | -12.8 | (-52.4, 59.5) | -14.3 | (-60.2, 84.6) | | |
| | BC | 3.6 | (-36.8, 69.6) | 0.5 | (-39.2, 66.2) | -23.0 | (-66.9, 79.4) | -14.3 | (-59.8, 83.0) | | |
| | DC | 6.0 | (-25.7, 51.4) | 7.9 | (-15.0, 36.9) | -3.4 | (-43.2, 64.5) | 2.5 | (-65.9, 208.3) | | |
| | UFP | 13.3 | (-31.2, 86.5) | 18.4 | (-15.0, 64.9) | 2.3 | (-53.5, 124.8) | -17.4 | (-62.1, 79.9) | | |
| | AMP | 12.4 | (-23.1, 64.4) | -2.9 | (-32.7, 40.0) | -23.1 | (-57.0, 37.5) | -26.7 | (-61.5, 39.4) | | |
| | $NO_2$ | -3.9 | (-38.8, 50.8) | 9.6 | (-28.4, 67.7) | -13.1 | (-58.0, 80.0) | -11.3 | (-55.6, 77.3) | | |
| | $O_3$ | 0.5 | (-20.5, 27.0) | -11.2 | (-33.7, 19.0) | -13.7 | (-48.1, 43.4) | -15.3 | (-49.3, 41.5) | | |
| | CO | -6.6 | (-43.7, 55.1) | 3.6 | (-38.6, 74.7) | -14.8 | (-64.7, 105.6) | -1.8 | (-55.0, 114.6) | | |
| | $SO_2$ | 7.2 | (-37.5, 83.9) | 6.6 | (-35.5, 76.2) | -24.4 | (-57.9, 35.8) | -9.7 | (-24.7, 8.3) | | |
| B. Spike protein IgG (N=74) | $PM_{2.5}$ | 25.0 | (-4.3, 63.2) | 32.7 | (-6.5, 88.5) | -5.8 | (-28.2, 23.6) | 15.6 | (-22.0, 71.3) | 34.9 | (2.1, 78.1) |
| | BC | 42.0 | (-16.2, 140.5) | 42.0 | (-21.5, 157.0) | 7.7 | (-33.0, 73.1) | 20.8 | (-28.3, 103.3) | 42.6 | (-10.5, 127.1) |
| | DC | 26.3 | (-16.0, 89.9) | 15.4 | (-12.8, 52.8) | -4.7 | (-28.1, 26.5) | 7.0 | (-44.3, 105.7) | 10.3 | (-25.3, 62.8) |
| | UFP | -10.4 | (-43.8, 42.6) | 20.6 | (-13.5, 68.0) | 21.5 | (-15.1, 73.8) | 15.6 | (-20.2, 67.6) | -14.2 | (-50.4, 48.5) |
| | AMP | 29.6 | (-10.2, 87.0) | 30.3 | (-12.3, 93.6) | -13.5 | (-43.2, 31.7) | -10.4 | (-48.0, 54.3) | 6.8 | (-27.5, 57.3) |
| | $NO_2$ | 16.4 | (-27.7, 87.4) | 25.2 | (-24.3, 107.0) | 7.7 | (-27.1, 59.2) | 20.0 | (-24.5, 90.7) | 20.7 | (-25.5, 95.5) |
| | $O_3$ | -2.3 | (-23.6, 25.0) | -22.4 | (-44.8, 8.9) | -13.6 | (-33.7, 12.5) | -11.0 | (-33.5, 19.2) | -1.1 | (-37.2, 55.7) |
| | CO | 28.4 | (-26.4, 124.0) | 43.0 | (-22.6, 164.3) | 21.0 | (-23.1, 90.4) | 26.7 | (-20.2, 101.0) | 53.5 | (-11.5, 166.2) |
| | $SO_2$ | 5.2 | (-37.0, 75.7) | -6.9 | (-46.4, 61.6) | -17.8 | (-39.3, 11.2) | -4.2 | (-14.0, 6.7) | 12.4 | (-15.2, 49.1) |

$PM_{2.5}$, BC and DC are measured in μg/m³, UFP and AMP in particles/cm³, $NO_2$, $SO_2$ and CO in ppb and $O_3$ in ppm.

**Table 5. Gas stove use and median antibody levels.**

| | Pollution day | Day 1–14 | Day 15–28 | Day 29–42 | Day 43–56 | Day 57–90 |
|---|---|---|---|---|---|---|
| | Blood draw day | Day 15 | Day 29 | Day 43 | Day 57 | Day 90 |
| **Log-transformed value** | | | | | | |
| **Spike protein IgG** | Yes | 11.1 | 12.5 | 14.6 | 14.2 | 13.7 |
| | No | 11.0 | 11.6 | 14.6 | 14.6 | 14.0 |
| **Neutralizing antibody** | Yes | 4.3 | 4.3 | 7.8 | 7.3 | |
| | No | 4.3 | 4.3 | 7.6 | 8.0 | |
| **Real value** | | | | | | |
| **Spike protein IgG (AU/ml)** | Yes | 2,134 | 5,749.5 | 24,651 | 18,674 | 13,409 |
| | No | 2,058.5 | 3,136.5 | 19,477 | 18,132 | 15,869 |
| **Neutralizing antibody** | Yes | 20 | 20 | 221 | 157 | |
| | No | 20 | 20 | 193 | 249 | |

difference between the median value of Neutralizing Ab or spike protein IgG in participants with gas stoves compared to those without (Table 5).

## Discussion

In this pilot study, we assessed the impact of exposure to air pollutants on the serologic immune responses to vaccination against SARS-CoV-2 in adults. We observed a non-statistically significant decrease in serum NAb titers associated with increased pollution concentrations following the second vaccine. In contrast, non-statistically significant increases in spike protein IgG concentrations were associated with an increase in several pollutants at early timepoints after the first vaccine but also at the 90-day time point. When considering the type of pollution, there did not appear to be different effects of particulate ($PM_{2.5}$, BC, DC, UFP and AMP) compared to gaseous pollutants ($NO_2$, $SO_2$ and $O_3$) regarding the immune response to vaccination.

The non-statistically significant 13–23% reduction in NAb titers associated with $PM_{2.5}$ and BC concentrations we observed is on a similar scale as the 23% reduction in NAb associated with an increase in a composite measure of air pollution observed in the Zhang (2022) study of the SINOVΛC vaccine. Of note, the composite measure of air pollution exposure (including multiple individual pollutants, estimated inhalation rate and outdoor exposure time) used in the SINOVΛC study is not easily comparable to our central monitoring site estimates of individual pollutants (e.g., $PM_{2.5}$, BC). In addition to the difficulty of direct comparison between studies, the clinical significance of the reduction in NAb associated with increased air pollution in both our study and the SINOVΛC study is unclear. A precise threshold of antibody that confers protection from COVID remains undefined but it appears that as antibody wanes, infection rates rise and boosting antibody provides improved protection [22]. Thus, it is reasonable to hypothesize that lower antibody titers will diminish vaccine efficacy. Though the findings of our small study were not statistically significant, given the consistent trend across multiple pollutants and potential practical implications, this topic deserves further study. If the efficacy of the vaccine is compromised by increased concentrations of air pollution, potentially a higher vaccine doses or additional booster doses may be beneficial.

Secondly, non-statistically significant increases in spike protein IgG titers were associated with an increase in several pollutants for early timepoints after the first vaccine but also at the 90-day time point (63 days from the second vaccine). As part of the COVICAT study in Spain, Kogevinas et al. [15] focused on the immune response after vaccination with several vaccines including AZD1222 and the Moderna and Pfizer RNA SARS-CoV-2 vaccines. This study observed around a 5–10% reduction in IgM (one month after vaccination) or IgG (anytime) to spike antigen for IQR increases in $PM_{2.5}$, black carbon and $NO_2$. This study also observed a more rapid drop in IgG in those exposed to a higher concentration of

particulate pollution [15]. In contrast to their findings, our study observed an increase in spike protein IgG in the early (first 29 days) and late period (57–90) after vaccination, albeit in a smaller cohort. Our study is difficult to compare to COVICAT given the different parent vaccines studies, populations and pollution measures. In addition to assessing longer term exposure (1 year prior to vaccination), the COVICAT study relied on corrected estimates of $PM_{2.5}$, $NO_2$ and $O_3$ data from 2010 and a separate estimated model for BC. In contrast, our study focused on one vaccine (AZD1222) and had real-time measurements of a wide variety of particulate and gaseous pollutants in the two-week period prior to each blood draw. While the sample size of the COVICAT study was larger, our population were only vaccinated with a single type of COVID-19 vaccine. Our group were all studied early in the pandemic, and we excluded any participants who had prior infection. In this way, we were able to assess the association between air pollution and the immune response to the AZ SARS-CoV-2 vaccine without the complication of natural immunity from infection.

The timing of air pollution exposure related to the first or second vaccine also appeared to be important. The largest reductions in NAb were observed at the dates furthest from the second vaccine (i.e., day 57). In contrast, increased concentrations of air pollution were associated with increases in spike protein IgG after the first (Day 15, 29) and second (Days 90) vaccinations. Past observations of impaired toll-like receptor activation and impaired antiviral immune cell signaling associated with air pollution and other inhaled toxins may also contribute to the mechanisms underpinning this observation of decreased NAb [13]. As MHC-1/CD8 + T lymphocyte signaling is involved in immunity from adenovirus vector vaccines [23], the finding of increased spike protein IgG associated with increases in $PM_{2.5}$ and black carbon may be related to particulate matter enhancement of cytotoxic of CD8 + T lymphocyte activity [24]. The clinical relevance of the apparent difference of the spike protein IgG and NAb responses to air pollution will require further study.

During the early stage of the COVID-19 pandemic (2020), many individuals remained in their home due, in part, to the reduction in travel outside of work. Participants in the present study spent roughly 20 hours on average at home each day. While this time at home helped reduce exposure misclassification when applying outdoor ambient air pollution estimates, it led an increased duration of indoor exposure. As we did not observe a consistent difference in NAb or spike protein IgG in those living in home with gas stoves vs. those without gas stoves, the role of indoor air pollution exposures on antibody response to SARS-CoV-2 vaccine will likely require more carefully controlled exposure studies to better characterize this relationship.

Our study is unique in that we were able to assess serial antibody measurements in the same non-infected population who received a single type of COVID-19 vaccine and for whom robust pollutant data was available in the two weeks immediately prior to each lab draw. However, there are several limitations to consider. The primary limitation of our pilot study was a reduced statistical power due to the small sample size. While exclusion of participants living more than 15 miles of the central monitor reduced the number of eligible participants, this exclusion should have served to reduce exposure misclassification. The exposure misclassification in our study would likely be non-differential with respect to vaccine response and would be expected to attenuate the estimated pollution/vaccine response associations towards the null. Secondly, our observations were conducted in a low pollution setting. Though there is a robust literature of health effects in both low and high air pollution settings, this may limit generalizability to other populations who experience higher concentrations or different sources of air pollution. Furthermore, if a threshold exists for the immunotoxic effects of air pollutants, exposures in our study could fall below a threshold for effect. In addition to providing a larger sample, a larger multicenter study in the future would also add valuable variability to the exposure types and concentrations. Thirdly, we were also mindful of the potential for batching of participants all experiencing the same pollution. However, with the start dates of the vaccination series spanning 3 months, we viewed this risk of loss of temporal variability as minimal. Fourthly, we recognize that temporal confounding related to pandemic dynamics including viral circulation, changes in air pollution, and other behavioral changes in the population secondary to the COVID-19 pandemic were likely present. Specifically, viral circulation (community infection) would likely be correlated with decreased pollution (decreased car/plane travel) and increased Neutralizing antibody levels in participants if infected. This would theoretically lead to an underestimation of the

magnitude of the decreased antibody response to $PM_{2.5}$ (and other pollutants). Though ultimately, the risk of confounding from viral circulation was minimized by the cautious distancing approach undertaken by participants to avoid respiratory infections during the clinical trial minimized their risk of COVID-19 infection. Finally, the findings for the vectored vaccine from AZ (not available in the US) and findings for RNA vaccines that are now used in the United States may be different, limiting the generalizability of our study.

## Conclusion

In a small study of participants vaccinated by the COVID-19 vaccine AZD1222, the serologic response of NAb to vaccination may be negatively impacted by the effect of air pollution (including traffic related air pollution). Though not statistically significant, the association between air pollutants/NAb had a different pattern than air pollutants/spike protein IgG. The actual biological significance of any increase or decrease in the immune titers will need to be determined in future work. The trends observed in this study deserve further study in a larger multicenter analysis, and with other vaccines, as confirmation of these results could have practical implications for vaccine development and implementation.

## Supporting information

**S1 Fig.  A) Percent difference in neutralizing antibody (NAb) titers at days 15 and 29 after primary vaccination and B) days 43 and 57 after secondary vaccination, associated with an IQR increase in multiple pollutants in the previous 14 days. Of note, the upper CI for DC at day 57 extends past the end of the graph (denoted by a double forward slash).**
(TIF)

**S2 Fig.  A) Percent difference in spike antibody IgG concentrations at days 15 and 29 after primary vaccination and B) 43, 57 and 90 after secondary vaccination associated with an IQR increase in multiple pollutants in the previous 14 days. Of note, the upper CI for BC at day 29 and CO at days 29 and 90 extends past the end of the graph (denoted by a double forward slash).**
(TIF)

**S1 Text.  Interquartile Range (IQR) discussion.**
(DOCX)

## Acknowledgements

We appreciate Astra Zeneca making the data available from the Phase 3 trial of the ChAdOx1 vectored COVID vaccine (AZD1222) for the secondary analysis of this pilot study.

## Author contributions

**Conceptualization:** Daniel Croft, David Q. Rich, Sally W. Thurston, Todd A. Jusko, Edward E. Walsh, Ann R. Falsey.

**Data curation:** Daniel Croft, Carl J. Johnston, Angela R. Branche, Philip K. Hopke, Kelly Thevenet-Morrison, Md Rayhanul Islam, Catherine Bunce, Michael C. Keefer, Edward E. Walsh, Ann R. Falsey.

**Formal analysis:** Daniel Croft, Carl J. Johnston, Philip K. Hopke, Kelly Thevenet-Morrison, Sally W. Thurston, Todd A. Jusko, Edward E. Walsh, Ann R. Falsey.

**Funding acquisition:** Ann R. Falsey.

**Investigation:** Daniel Croft, David Q. Rich, Todd A. Jusko, Ann R. Falsey.

**Methodology:** Daniel Croft, David Q. Rich, Philip K. Hopke, Kelly Thevenet-Morrison, Sally W. Thurston, Todd A. Jusko, Edward E. Walsh, Ann R. Falsey.

**Project administration:** Daniel Croft, Carl J. Johnston, Kelly Thevenet-Morrison, Ann R. Falsey.

**Resources:** Daniel Croft, Ann R. Falsey.

**Software:** Daniel Croft, Kelly Thevenet-Morrison.

**Supervision:** Daniel Croft, David Q. Rich, Kelly Thevenet-Morrison, Sally W. Thurston, Edward E. Walsh, Ann R. Falsey.

**Validation:** Daniel Croft, Ann R. Falsey.

**Visualization:** Daniel Croft, Kelly Thevenet-Morrison, Todd A. Jusko, Md Rayhanul Islam, Ann R. Falsey.

**Writing – original draft:** Daniel Croft, Carl J. Johnston, Ann R. Falsey.

**Writing – review & editing:** Daniel Croft, Carl J. Johnston, Angela R. Branche, David Q. Rich, Philip K. Hopke, Kelly Thevenet-Morrison, Sally W. Thurston, Todd A. Jusko, Md Rayhanul Islam, Catherine Bunce, Michael C. Keefer, Edward E. Walsh, Ann R. Falsey.

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
