## [Decision Letter · Decision Letter 0]

22 Dec 2024

PGPH-D-24-00744

Ambient air pollution exposure and effects on neutralizing antibody titers following SARS-CoV-2 vaccination in adults

Dear Dr. Croft,

Thank you for submitting your manuscript to PLOS Global Public Health. After careful consideration, we feel that it has merit but does not fully meet PLOS Global Public Health’s publication criteria as it currently stands. Therefore, we invite you to submit a revised version of the manuscript that addresses the points raised during the review process.

The manuscript has been assessed by two reviewers, and their comments are provided below. The reviewers clarifications about the study design, and in particular several revisions were requested regarding the statistical assessments, including to avoid overstating non-significant results. Could you please revise the manuscript to address the concerns raised?

We look forward to receiving your revised manuscript.

Kind regards,

Marianne Clemence

Staff Editor

Journal Requirements:

 1. Please send a completed 'Competing Interests' statement, including any COIs declared by your co-authors. If you have no competing interests to declare, please state "The authors have declared that no competing interests exist". Otherwise please declare all competing interests beginning with the statement "I have read the journal's policy and the authors of this manuscript have the following competing interests:" 2. Please amend your detailed Financial Disclosure statement. This is published with the article. It must therefore be completed in full sentences and contain the exact wording you wish to be published. **Please only choose the relevant sentences from below** 1. Please clarify all sources of funding (financial or material support) for your study. List the grants (with grant number) or organizations (with url) that supported your study, including funding received from your institution. 2. State the initials, alongside each funding source, of each author to receive each grant.3. State what role the funders took in the study. If the funders had no role in your study, please state: “The funders had no role in study design, data collection and analysis, decision to publish, or preparation of the manuscript.”4. If any authors received a salary from any of your funders, please state which authors and which funders. If you did not receive any funding for this study, please simply state: “The authors received no specific funding for this work.” 3. In the online submission form, you indicated that " As the proprietary data on neutralizing antibody and spike protein IgG were obtained from Astra Zeneca for this study, data requests for NAb and spike protein IgG will need to be addressed to Astra Zeneca for permission to use these data.".  All PLOS journals now require all data underlying the findings described in their manuscript to be freely available to other researchers, either 1. In a public repository, 2. Within the manuscript itself, or 3. Uploaded as supplementary information. This policy applies to all data except where public deposition would breach compliance with the protocol approved by your research ethics board. If your data cannot be made publicly available for ethical or legal reasons (e.g., public availability would compromise patient privacy), please explain your reasons by return email and your exemption request will be escalated to the editor for approval. Your exemption request will be handled independently and will not hold up the peer review process, but will need to be resolved should your manuscript be accepted for publication. One of the Editorial team will then be in touch if there are any issues. 4. We do not publish any copyright or trademark symbols that usually accompany proprietary names, eg (R), (C), or TM  (e.g. next to drug or reagent names). Please remove all instances of trademark/copyright symbols throughout the text, including "Meso Scale Discovery®" on page 6. 5. Please provide an Author Summary. This should appear in your manuscript between the Abstract (if applicable) and the Introduction, and should be 150–200 words long. The aim should be to make your findings accessible to a wide audience that includes both scientists and non-scientists. Sample summaries can be found on our website under Submission Guidelines: https://journals.plos.org/globalpublichealth/s/submission-guidelines#loc-parts-of-a-submission 6. "AZ vaccine and air pollution (1).tif": Please confirm whether you drew the images / clip-art within the figure panels by hand. If you did not draw the images, please provide (a) a link to the source of the images or icons and their license / terms of use; or (b) written permission from the copyright holder to publish the images or icons under our CC-BY 4.0 license. Alternatively, you may replace the images with open source alternatives. See these open source resources you may use to replace images / clip-art:- https://commons.wikimedia.org-
https://openclipart.org/ 

Additional Editor Comments (if provided):

Reviewers' comments:

Reviewer's Responses to Questions

**Comments to the Author**

1. Does this manuscript meet PLOS Global Public Health’s publication criteria ? Is the manuscript technically sound, and do the data support the conclusions? The manuscript must describe methodologically and ethically rigorous research with conclusions that are appropriately drawn based on the data presented.

Reviewer #1: Partly

Reviewer #2: Partly

2. Has the statistical analysis been performed appropriately and rigorously?

Reviewer #1: Yes

Reviewer #2: No

3. Have the authors made all data underlying the findings in their manuscript fully available (please refer to the Data Availability Statement at the start of the manuscript PDF file)?

Reviewer #1: Yes

Reviewer #2: No

4. Is the manuscript presented in an intelligible fashion and written in standard English?

Reviewer #1: Yes

Reviewer #2: Yes

5. Review Comments to the Author

Reviewer #1: This interesting pilot study addresses an important research question: whether acute air pollution exposure impairs vaccine responses. If the data are confirmed in other settings, larger studies should be conducted to evaluate to what extent vaccination protects from the deleterious effects of air pollution on Covid incidence and severity.

The authors have made a great effort, but some aspects are unclear and need revaluation.

1. The authors need to explain why they did not perform the same analysis in the placebo group. The findings would have strengthened the results.

2. Temporal confounding may be an issue, as the authors did not control for that. Temporal changes in air pollution were possibly related to changes in viral circulation (which could affect the circulating levels of antibodies).

Minor comments.

The authors say in the introduction that the vaccination started in the winter 2020. I think it started later for the general population and this should be corrected.

Reviewer #2: The linear mixed-effects model used for the statistical analysis is well-suited to the study design and data, however there are a few aspects that require further explanations. These aspects include the sample size, the adjustment of covariates, and the evaluation of effect significance, in this case non-significant statistical results.

Regarding the results, the manuscript has to consistently communicate the non-statistically significant findings, particularly in the abstract and all related sections.

I also recommend the conclusion to state that no statistical significant effects were found. To be fair, there are such statements in the start of the Discussion and the conclusion says that association is "imprecise". This should be clear.

The sample was limited to the number of participants and, more importantly, the number of residents within 15 miles of the air pollution measuring station. These restrictions limited the sample size to 74 individuals. What can you say about a minimum sample size regarding a few criteria? How much power can you establish from the sample size in the study?

Another significant limitation of the study is that all participants reside in the same city, which typically experiences low levels of air pollution. Also, if air pollution increases even slightly for any given moment during the study period, all individuals receiving the vaccine at the same time are exposed to the variation of same level of air pollution.

In fact, the descriptive analysis indicates that air pollutant concentrations remained very stable throughout the study, making it more challenging to assess any effects.

I understand that variables sex and age were not included in all models due to not finding significant coefficients in the regression. However, these coefficients are not shown in the results.

The methodology says that direction and magnitude of the effects were considered, rather than significance. I understand the reasoning about direction, but it is not possible to say much about direction/magnitude when uncertainty intervals are large, including positive and negative values. Therefore, the statistical significance has to play a role. This should be clear in the methodology.

In this context, conducting a multi-center study would provide more valuable insights, benefiting from larger samples and more variability of exposures across participants. Therefore, the possibility of future multi-center studies should be indicated in the Discussion section.

About results:

- Table 3: Insert the information that values refer to the log-transformed neutralizing antibody levels in the second line.

- I understand that Figures 1 and 2 are suitable for visualizing the effects. However, if I am not mistaken, all information in Figure 1 and Figure 2 is already present in Table 4. If this is really the case, Figure 1 and Figure 2 are redundant and should be removed from the manuscript. In this way, a results description should be in Table 4 accordingly.

In the abstract and the Discussion, it is said that unexpectedly spike protein IgG increased 14 days after vaccination. Still, evaluations for most COVID-19 vaccines, including the Astra Zeneca vaccine, should consider a period of 14 days to start counting protection according to WHO guidelines. I would not say that this increase in spike protein IgG was unexpected. Therefore, I recommend avoiding such a statement.

Note: I did not receive the supplement indicated in line 167.

6. PLOS authors have the option to publish the peer review history of their article (what does this mean? ). If published, this will include your full peer review and any attached files.

**Do you want your identity to be public for this peer review?** For information about this choice, including consent withdrawal, please see our Privacy Policy .

Reviewer #1: **Yes: ** Francesco Forastiere

Reviewer #2: No

---

## [Decision Letter · Decision Letter 1]

25 Feb 2025

PGPH-D-24-00744R1

Ambient air pollution exposure and effects on neutralizing antibody titers following SARS-CoV-2 vaccination in adults

Dear Dr. Croft,

Thank you for submitting your manuscript to PLOS Global Public Health. After careful consideration, we feel that it has merit but does not fully meet PLOS Global Public Health’s publication criteria as it currently stands. Therefore, we invite you to submit a revised version of the manuscript that addresses the points raised during the review process.

Reviewer 2 has recommended additional minor revisions to improve your manuscript, including data presentation and clarity of specific sentences. Please see their comments below and address all in your revised submission.

We look forward to receiving your revised manuscript.

Kind regards,

Jennifer Tucker, PhD

Staff Editor

Journal Requirements:

Reviewers' comments:

Reviewer's Responses to Questions

**Comments to the Author**

1. If the authors have adequately addressed your comments raised in a previous round of review and you feel that this manuscript is now acceptable for publication, you may indicate that here to bypass the “Comments to the Author” section, enter your conflict of interest statement in the “Confidential to Editor” section, and submit your "Accept" recommendation.

Reviewer #1: All comments have been addressed

Reviewer #2: (No Response)

2. Does this manuscript meet PLOS Global Public Health’s publication criteria ? Is the manuscript technically sound, and do the data support the conclusions? The manuscript must describe methodologically and ethically rigorous research with conclusions that are appropriately drawn based on the data presented.

Reviewer #1: Yes

Reviewer #2: Yes

3. Has the statistical analysis been performed appropriately and rigorously?

Reviewer #1: Yes

Reviewer #2: Yes

4. Have the authors made all data underlying the findings in their manuscript fully available (please refer to the Data Availability Statement at the start of the manuscript PDF file)?

Reviewer #1: Yes

Reviewer #2: (No Response)

5. Is the manuscript presented in an intelligible fashion and written in standard English?

Reviewer #1: Yes

Reviewer #2: Yes

6. Review Comments to the Author

Reviewer #1: The authors have provided acceptable responses to the critiques

Reviewer #2: In the Author Summary, a suggestion is to rephrase in line 53 to: the spike protein, a selected antigen from the SARS-CoV-2 virus.

This sentence requires correction - it seems truncated: “in a simple descriptive sensitivity analysis, we explored whether or not median antibody concentrations different in individuals with or without gas stoves (without modelling ambient air pollution).”

In my opinion, the tables are more informative instead of Figures 1 and 2 for the reasons that the authors indicated, i.e., that any reader can review the actual data values from the tables. I highly suggest to have the tables in the main manuscript instead of the Figures.

Some units are missing from the information in Tables 2, 3, and 5.

7. PLOS authors have the option to publish the peer review history of their article (what does this mean? ). If published, this will include your full peer review and any attached files.

**Do you want your identity to be public for this peer review?** For information about this choice, including consent withdrawal, please see our Privacy Policy .

Reviewer #1: No

Reviewer #2: No

---

## [Decision Letter · Decision Letter 2]

10 Apr 2025

PGPH-D-24-00744R2

Ambient air pollution exposure and effects on neutralizing antibody titers following SARS-CoV-2 vaccination in adults

Dear Dr. Croft,

Thank you for submitting your manuscript to PLOS Global Public Health. After careful consideration, we feel that it has merit but does not fully meet PLOS Global Public Health’s publication criteria as it currently stands. Therefore, we invite you to submit a revised version of the manuscript that addresses the points raised during the review process.

Per Reviewer 2's comment, please ensure that the new Figure 1 is available for review.

We look forward to receiving your revised manuscript.

Kind regards,

Mara Jana Broadhurst, M.D., Ph.D.

Academic Editor

Journal Requirements:

Additional Editor Comments (if provided):

Reviewers' comments:

Reviewer's Responses to Questions

**Comments to the Author**

1. If the authors have adequately addressed your comments raised in a previous round of review and you feel that this manuscript is now acceptable for publication, you may indicate that here to bypass the “Comments to the Author” section, enter your conflict of interest statement in the “Confidential to Editor” section, and submit your "Accept" recommendation.

Reviewer #1: All comments have been addressed

Reviewer #2: (No Response)

2. Does this manuscript meet PLOS Global Public Health’s publication criteria ? Is the manuscript technically sound, and do the data support the conclusions? The manuscript must describe methodologically and ethically rigorous research with conclusions that are appropriately drawn based on the data presented.

Reviewer #1: Yes

Reviewer #2: Yes

3. Has the statistical analysis been performed appropriately and rigorously?

Reviewer #1: Yes

Reviewer #2: Yes

4. Have the authors made all data underlying the findings in their manuscript fully available (please refer to the Data Availability Statement at the start of the manuscript PDF file)?

Reviewer #1: Yes

Reviewer #2: Yes

5. Is the manuscript presented in an intelligible fashion and written in standard English?

Reviewer #1: Yes

Reviewer #2: Yes

6. Review Comments to the Author

Reviewer #1: The authors have provided a complete response to the comments

Reviewer #2: All comments have been addressed. However, the manuscript contains a reference to Figure 1, which is missing from the submission and was not present in previously submitted versions. I went through the submission file multiple times but could not locate it. Please provide this figure.

"Figure 1: Design of the pilot secondary analysis of a clinical trial, with results."

7. PLOS authors have the option to publish the peer review history of their article (what does this mean? ). If published, this will include your full peer review and any attached files.

**Do you want your identity to be public for this peer review?** For information about this choice, including consent withdrawal, please see our Privacy Policy .

Reviewer #1: No

Reviewer #2: No

---

## [Editor Report · Decision Letter 3]

17 Apr 2025

Ambient air pollution exposure and effects on neutralizing antibody titers following SARS-CoV-2 vaccination in adults

PGPH-D-24-00744R3

Dear Dr. Croft,

We are pleased to inform you that your manuscript 'Ambient air pollution exposure and effects on neutralizing antibody titers following SARS-CoV-2 vaccination in adults' has been provisionally accepted for publication in PLOS Global Public Health.

Best regards,

Mara Jana Broadhurst, M.D., Ph.D.

Academic Editor
